

# An open-access CMIP5 pattern library for temperature and precipitation: Description and methodology.

Cary Lynch[1], Corinne Hartin[1], Ben Bond-Lamberty[1], and Ben Kravitz[2]

[1]Pacific Northwest National Laboratory, Joint Global Change Research Institute, 5825 University Research Court, Suite 3500, College Park, MD 20740
[2]Atmospheric Sciences and Global Change Division, Pacific Northwest National Laboratory, 902 Battelle Boulevard, Richland, WA 99352

*Correspondence to:* Cary Lynch (cary.lynch@pnnl.gov)

**Abstract.** Pattern scaling is used to efficiently emulate general circulation models and explore uncertainty in climate projections under multiple forcing scenarios. Pattern scaling methods assume that local climate changes scale with a global mean temperature increase, allowing for spatial patterns to be generated for multiple models for any future emission scenario. Of the possible techniques used to generate patterns, the two most prominent are the delta and least squared regression methods. For uncertainty quantification and probabilistic statistical analysis, a library of patterns with descriptive statistics for each file would be beneficial, but such a library does not presently exist. This paper presents patterns from all CMIP5 models for temperature and precipitation on an annual and sub-annual basis, along with the code used to generate these patterns. We explore the differences and statistical significance between patterns generated by each method and assess performance of the generated patterns across methods and scenarios. Regardless of epoch chosen, local temperature sensitivity to global mean temperature change is similar with differences of $\leq 0.2^{\circ}$C. Differences in patterns across seasons between methods and epochs were largest in high latitudes (60-90$^{o}$N/S). Bias and mean errors between modeled and pattern predicted output from the linear regression method were smaller than patterns generated by the delta method. Across scenarios, differences in the linear regression method patterns were more statistically significant, especially at high latitudes. We found that pattern generation methodologies were able to approximate the forced signal of change to within $\leq 0.5^{\circ}$C, but choice of pattern generation methodology for pattern scaling purposes should be informed by user goals and criteria. The dataset and netCDF data generation code are available at http://doi.org/10.5281/zenodo.235905.

## 1 Introduction

Projections of future climate are bound by a limited number of possible forcing scenarios, making the task of robustly exploring uncertainties in climate projections difficult. In the absence of a large sample of model experiments to draw upon, extrapolation from and interpolation between scenarios can be used to reduce uncertainty from future forcing by spanning a much larger range of emission scenarios (Santer et al., 1990; Dessai et al., 2005). One such computationally efficient method to emulate many different future forcing scenarios scaled from general circulation models (GCMs) is *pattern scaling*.



Pattern scaling was initially established to enable the creation of transient climate projections from the steady state response of a GCM to a doubling of the preindustrial $CO_2$ concentration (Santer et al., 1990). Under the assumption that a climate variable from a GCM scales proportionally with global mean temperature (GMT) change, patterns are derived from multiple GCMs. Those patterns can then be scaled in magnitude by a specified GMT change or by the GMT change obtained from a

simple climate model (SCM) to span a wide range of future scenarios (Moss et al., 2010) that have not been simulated by full GCMs (Figure 1).

Pattern estimation methodologies have evolved into two general types. The first and most common is the time-slice method or the "delta" method, where local future change in a climate variable is normalized by future GMT change averaged over some chosen time period, hereafter referred to as an *epoch* (Osborn et al., 2009; Tebaldi and Arblaster, 2014; Herger et al., 2015).

The second is the linear regression method, which uses ordinary least squares regression coefficients to fit local trends to a GMT time series (Mitchell, 2003; Ruosteenoja et al., 2007; Lopez et al., 2013; Osborn et al., 2015). In terms of computational efficiency, the delta method is the fastest, which is a reason why it is predominantly used (Herger et al., 2015). However, in terms of skill in trend estimation and adaptability of additional predictors, the linear regression method is preferable (Mitchell, 2003; Frieler et al., 2012; Lustenberger et al., 2014; Barnes and Barnes, 2015).

In pattern scaling several broad assumptions are made. First, it is assumed that patterns generated under different forcing scenarios are not significantly different. Tebaldi and Arblaster (2014) found that patterns from different scenarios were highly spatially correlated, and that choice of scenario did not explain a significant proportion of variability in patterns when using the delta pattern scaling method. For the linear regression methodology, Mitchell (2003) found that the linear regression method reduces the influence of the non-linearities arising from differing rates of warming, but the magnitudes of estimation errors

vary between scenarios and for other climate variables due to non-linearity in GMT relationships.

Second, it is assumed that responses to external forcing and internal variability are independent, implying that anthropogenic forcings do not modify the internal variability of the climate system (Mitchell, 2003; Lopez et al., 2013), but this premise is not always true (Screen, 2014). Changes in variability may introduce estimation errors in pattern fit; in practice, estimation errors introduced through this assumption at the global scale were small, although they can be large enough at the regional scale to

mislead adaptation decisions (Lopez et al., 2013).

Third, it is assumed that local change scales proportionally with GMT change, and that the relationship is stationary over time (Mitchell, 2003). This assumption is not always true in the climate system, especially considering different forcing scenarios and spatial heterogeneity of projected change. For temperature variables, the assumption of stationarity is generally valid, but the magnitudes of estimation errors vary between scenarios for non-temperature variables (Frieler et al., 2012) and temperature

extremes on the upper tail of the temperature distribution (Lustenberger et al., 2014). Lopez et al. (2013) found that when pattern scaling temperature extremes over Southern Europe, the magnitude of the error in the pattern estimates was large. In linear regression, only the error term is assumed to have a normal distribution, so it is likely that climate extremes would yield high error terms. This can be problematic when constructing confidence intervals but is not necessarily a limitation in the pattern scaling methodology, nor in the resulting patterns (Lustenberger et al., 2014).



Currently there are three prominent online tools and software that generate climate patterns and scale them with specific SCM scalers (Wigley, 2008; Osborn et al., 2009; Castruccio et al., 2014). These tools lock the user into using a specific SCM and do not provide pattern data and diagnostics, which can be important for understanding individual model scaled patterns. For uncertainty quantification and probabilistic statistical analysis, a library of patterns with descriptive statistics for each file would be beneficial, but such a library does not presently exist.

This paper presents both a description of our data library and our methodology to generate these patterns. We focus on spatial heterogenity and magnitude of the local response to GMT change (hereafter called "pattern sensitivity") from both pattern estimation methodologies. We also explore the assumption that the patterns were consistent across multiple forcing scenarios.

## 2   Data/Methods

### 2.1   Climate Models

We employed three sets of experiments from the Coupled Model Intercomparison Project Phase 5 (CMIP5; Taylor et al., 2011). The 'historical' experiment was used to construct reference epochs for use in pattern creation via the delta method. Model historical runs varied in length, so we used 1861 as the start of the historical period, and 2005 as the end. For future projections, we used the high-forcing rcp8.5 scenario, in which radiative forcing increases to 8.5 W/m$^2$ through the 21$^{st}$ Century (Riahi et al., 2011), and the mid-forcing rcp4.5 scenario, in which radiative forcing increases to 4.5 W/m$^2$ through the 21$^{st}$ Century (Thomson et al., 2011) achieved by limiting future emissions. For the future simulations, the start year was 2006, and the end year was 2100.

For assessment of patterns we chose a subset of twelve climate models (Table 1) from the 41 available models that included a historical and rcp4.5/rcp8.5 experiments. We used the first realization from each model and chose not to average over multiple realizations from each model. For this study, only seasonal and annual surface air temperature were analyzed, and in some cases, the influence of the land-ocean contrast was examined. For such cases, a land mask was applied using each model's native resolution, with 100% of the grid cell either land or ocean. Assessment of patterns generated by each methodology was done by examining the multi-model ensemble mean as well as the uncertainty represented by the model spread.

All model output was regridded to the lowest spatial resolution of the multi-model ensemble prior to the calculation of an ensemble mean or median. This was done for averaging purposes, as each model had a different spatial resolution. Regridding to the lowest resolution of the multi-model ensemble is a conservative assumption that avoids interpolation errors if the results are used for purposes of impact assessments.



## 2.2 Data Analysis

The delta pattern (DP) is described as follows:

$$DP_{MS} = \frac{\Delta TL_{MS}}{\Delta TG_{MS}} \tag{1}$$

For each model ($M$) and future scenario ($S$), local (2 dimensional, a value for each latitude and longitude pair) temperature change ($TL$) is normalized by global (scalar value) mean temperature change ($TG$), with respect to a 30 year reference epoch from the CMIP5 historical simulation.

All epochs were thirty years in length, as it was assumed that the length of epochs used should not alter the resulting pattern. Barnes and Barnes (2015) argue that the ideal epoch length is dependent on minimizing variable variance by selecting a epoch length with a high signal-to-noise ratio, which is largely dependent on length of time series, and whether the trend in the time series is linear. They found that for temperature, one-third the length of the time series is ideal, and for a 100 year time series (or longer) a thirty year epoch length is sufficient. Throughout the IPCC Fifth Assessment Report (AR5; Stocker et al., 2013) a 30-year reference epoch of 1986-2005 was used for discussion of projected anomalies; previous assessment reports used earlier epochs. In impact studies, a later reference epoch is more suitable because it is more representative of the current climate, and hence what socio-economic systems were already somewhat adapted to (Fowler et al., 2007; Herger et al., 2015). In adaptation/mitigation analyses, a pre-industrial control simulation epoch is often used as the baseline from which change is diagnosed, as this period is likely to provide the largest deviation from projected future climate, but for pattern generation, an epoch in the later half of the $20^{th}$ Century is often used (Osborn et al., 2009; Tebaldi and Arblaster, 2014).

For the aforementioned epoch variations, we used two reference epochs to generate patterns: a late $19^{th}$ (1861-1990) and a late $20^{th}$ (1971-2000) Century average, hereafter referred to as L19C and L20C respectively. The bulk of the epoch patterns use a future epoch that spans the last 30 years of the $21^{st}$ Century (2071-2100), but a mid $21^{st}$ Century (2041-2070) epoch was also used when examining epoch pattern differences. These epochs were hereafter referred to as L21C and M21C, respectively, in the figures and text.

The least squared regression (LSR) patterns were calculated from future forcing scenarios only. We use a least squares approach, which provides the best fit for calculating the regression pattern:

$$TL_{MS} = \alpha_{MS} + \beta_{MS} * TG_{MS} + \epsilon_{MS} \tag{2}$$

In this equation, $TG$ is the GMT time series (one-dimensional, unsmoothed), and $TL$ is the gridded time series (three dimensional). $\beta$ is a two-dimensional field of regression slopes, and $\epsilon$ is a two-dimensional residual term (error) stemming from linearly fitting the dependent variable to the predictor. $\alpha$ is the $y$-intercept, which we take to be 0 by only computing change, not absolute temperature.

To examine the assumption that the multi-model ensemble probability distribution and the sample mean between patterns and scenarios generated by each method were not significantly different, we calculated the Student's t-distribution probability.





This was done because the ensemble consists of only twelve models, which poorly samples the space of possible modeled climate realizations, and because we assume the ensemble variance for each pattern is the same. The resulting probability indicates where there is a significant difference between patterns generated by each method.

Pattern estimation can be skewed by local variability because large variability can mask the local warming signal. To identify areas where pattern fit is poor due to high variability, we calculated the detrended $21^{st}$ Century variance and the signal-to-noise ratio as defined by Hawkins and Sutton (2012). The signal-to-noise ratio identifies regions where the magnitude of the warming signal in relation to historical variability is large. The signal calculation in the signal-to-noise ratio makes an assumption that local temperature changes scale with global temperature (Hawkins and Sutton, 2012), similar to pattern scaling methodologies.

Performance metrics for pattern scaling across methodologies is difficult. For this study we quantified the differences between the reconstruction $\hat{B}$ and the actual model output $B$ via the root mean square error (RMSE) over the area-weighted difference at the end of the $21^{st}$ Century. In this instance RMSE is used to describe how well the predicted pattern emulates the actual model change, with lower RMSE indicating that the predicted pattern better captures the actual model change.

$$RMSE = \frac{\sqrt{\sum_x \left[ \left( \hat{B}(x) - B(x) \right) \cdot A(x) \right]^2}}{\sqrt{\sum_x \left[ A(x) \right]^2}} \tag{3}$$

where $A(x)$ is the area of the grid box $x$ and sums were calculated over all $x$.

# 3 Pattern Results

## 3.1 Pattern Differences

For the delta methodology, choice of epoch can be important, and in our ensemble, at the local spatial scale, absolute temperature differences between reference epochs were small, but differences in future epochs often exceeded $2^oC$ in rcp8.5, particularly over land and at high latitudes (Figure 2). Differences in variance across epochs were also small (Figure 3), and these relatively small differences in variance between epochs were not likely to affect the resulting temperature patterns. This may not be true when using other climate variables like precipitation, which may have large year to year or decadal natural variability in the observed period.

Patterns across epochs were similar despite differences in rate of GMT change and absolute temperature differences in epochs (Figure 4). Differences between reference epoch patterns were largest in the Northern Hemisphere mid and high latitudes, but differences were generally not significant, except for the Great Lakes region of North America in December through February (DJF) and over the North Pacific Basin and Eastern Asia in June through July (JJA). These differences in patterns across reference epochs were amplified when the mid $21^{st}$ Century future epoch is used, as compared to the late $21^{st}$ Century future epoch.

Regardless of epoch chosen for the delta method, the resulting patterns were similar to the regression patterns (Figure 5). The key idea in either pattern scaling method is that local temperature change scales with global temperature change, despite





different ways of calculating the local/global relationship. With the exception of the high latitudes, the differences in the annual pattern were small ($< 0.2°C$). Pattern differences were similar across seasons, but differences in patterns were the largest in DJF, particularly for the delta pattern with the earlier reference period. The regression patterns have a stronger temperature sensitivity to GMT change in the Northern Hemisphere and a weaker temperature sensitivity in the Southern Hemisphere at

high latitudes as compared to the delta methods. These differences in sensitivity stem from how each methodology capture the effect of Arctic amplification, where the warming trend in the Arctic is almost twice as large as the trend in the global average, but the effect of Arctic amplification on pattern generation is not explored in here.

There were few regions where the patterns differ significantly (Figure 5), and there were fewer significant differences between the regression method and the delta method using the L20C epoch over the L19C epoch. Significant differences between

patterns generated from each method were shown in the Baltic/ N. European region for both epochs in the annual and DJF pattern, but in the earlier epoch, significant differences across seasons were shown in the Northwest Pacific region. In general, the temperature patterns across methods were very similar.

To evaluate performance of each pattern methodology, accuracy was based on how well the patterns approximated the linear GMT change of $1^o$C simulated by each GCM. For this evaluation of metric the delta patterns largely underestimate the spatial

pattern, particularly over land and mid-high Northern latitudes (Figure 6). The Antarctic region is both overestimated (L21C/ L19C pattern) and underestimated (L21C/ L20C pattern) by a magnitude of $\geq 0.15°C$, which is generally larger than the error in the regression pattern estimates. Also, as shown in Figure 4, the delta patterns have a strong temperature sensitivity over the the Baltic/ N. European region. Overall, it appears that the regression pattern scaling method underestimates the relationship between global temperature and local temperature, but the degree to which it overestimates the relationship is small ($< 0.08°C$).

Emulator performance was also approximated by examining the RMSE between the actual and pattern predicted anomaly (Table 2). For this metric, the regression patterns also outperforms the delta pattens regardless of epoch. DJF RMSE were higher than the JJA RMSE, and the rcp4.5 RMSE was consistently lower than the rcp8.5 across methods. This may be because the rcp8.5 patterns largely underestimates the relationship between global and local temperature as seen in Figure 6. Nevertheless, Table 2 indicates that the both methodologies do well emulating actual model output.

Overall, the annual and seasonal patterns from each method were not significantly different from each other, regardless of reference epoch for the delta method. The differences were slightly larger when using an earlier reference epoch, but the regions where the ensemble differences were significant (above the 95% significance level) were small. Our small ensemble size (12 models with only one realization) may have contributed to lack of significance in differences across epoch patterns, particularly when using parametric tests like calculating p-values for the Student's t-test. A more robust analysis would include

multiple realizations from all available models.

### 3.2 Scenario Differences

To test the assumption that local temperature sensitivity to global mean temperature change, regardless of methodology (May, 2011), is consistent across scenarios we compare $21^{st}$ Century GMT signals by epoch differences and linear trend over the $21^{st}$ Century (Figure 7). For the rcp8.5 scenario, the GMT signals were very similar, despite the a larger ensemble spread in



the calculated linear trend. In the rcp4.5 scenario, the differences between the two ensemble mean GMT changes were as much as $1^oC$, which suggests that the way the global signal is calculated and the rate at which the signal changes plays a key role in understanding the differences between methods across scenarios. This is further supported by Mitchell (2003) who found that the GMT rate of change can have a significant impact on response patterns.

There were significant differences between patterns generated across scenarios, and the resulting pattern differed by more than $0.5°C$ in some regions (Figure 8). For the delta patterns, the largest differences across scenarios were in the Northern Hemisphere at high latitudes, areas where temperature variability is large (Figure 9). The differences in patterns generated by the regression method under different forcing scenarios were generally larger with statistically significant differences in the mid-high latitudes, particularly in the Arctic, land areas bordering the Mediterranean, and the subtropical South Pacific. The

rcp4.5 also has a lower signal-to-noise ratio than the rcp8.5 (Figure 8), which makes the pattern for the rcp4.5 scenario more difficult to estimate because the signal is harder to distinguish from the noise in this scenario.

Temperature change at high latitudes cannot be approximated by a linear relationship due to strong regional feedbacks, for example Arctic Amplification (Holland and Bitz, 2003), and therefore is not well predicted when using pattern scaling methods. The differences between scenarios are larger in the regression method, but both methods show similar spatial patterns.

To further examine why the regression method produces larger differences across scenarios, we looked at the linear fit of local temperature to GMT (Figure 10). In the rcp8.5 scenario, the $R^2$ values were large, but in the rcp4.5 scenario, $R^2$ values were much lower particularly along the Antarctic continent and in the North Atlantic. Even though the global/local fit is poorer in the rcp4.5 scenario, the lower forcing scenario predicted pattern is more like the actual model output (Table 2).

Large differences in patterns across scenarios were mainly due to a larger local/global ratio at high latitudes in the rcp4.5

scenario as compared to the rcp8.5 scenario despite lower local and global trends (Figure 11). These differences at high latitudes result from a steep temperature change gradient and the fast rate of change after sea/land ice has melted. Sensitivity of high latitudes to even small changes in GMT is evident across scenarios, but the rcp4.5 scenario overestimates this relationship, resulting in substantial differences in patterns between the scenarios, particularly for the regression methodology.

Differences between patterns across scenarios is further examined by separating the land and ocean patterns (Figure 12).

The differences between scenarios for the regression method when isolating the land/ocean pattern were comparatively large, especially over the Arctic and Antarctic regions. For the regression method, the rcp4.5 ocean only pattern sensitivity is $\geq 0.5°C$ than the rcp8.5 (ocean only) pattern sensitivity over the Arctic, and the rcp4.5 land only pattern sensitivity is $\geq 0.5°C$ than the rcp8.5 (land only) pattern sensitivity over the Antarctic. The differences in patterns across scenarios for the delta method when isolating the land/ocean pattern were small except over the Arctic region, which shows strong seasonal differences ($\geq 0.5°C$)

in boreal autumn (SON). In this way the delta method is more consistent across future forcing scenarios, which should be taken into consideration when choosing methodology.

Differences in patterns across scenarios were not surprising as pattern scaling has been shown to be less accurate for scenarios with stronger mitigation (May, 2011; Ishizaki et al., 2012), even though we found that for the models and scenarios we used, the lower forcing scenario patterns better emulated the actual model response. A weaker GMT signal coupled with non-linear

relationships between GMT change and local climate change (particularly in the Arctic) under strong mitigation scenarios



result in larger pattern errors. We found that differences in patterns between scenarios are more evident in the regression method as compared to the delta method, but similar features appear in the patterns produced by the delta method. How models incorporate sea-ice may also add to the variability of patterns across models, but this is a subject we have not explored.

## 4 Conclusions

The differences in patterns generated by each method were minor except at Northern Hemisphere high latitudes and along the Antarctic margin. The local to global fit is strong, and with the assumption of linearity, the regression methodology pattern outperforms the delta methodology pattern. The regression methodology patterns also have lower RMSE, and better emulates actual model output. The simplistic design of the regression method allows for additional predictors in the pattern equation and confidence intervals to be easily calculated. The delta method introduces complexity in choice of reference epoch and length of reference epoch, but we have found little difference between patterns across epochs.

Choice of scenario can affect the resulting pattern, particularly at high latitudes. With the regression methodology pattern, the GMT temperature sensitivity is stronger when using the rcp4.5 scenario because the GMT trend is proportionally smaller and changes in GMT have a stronger effect on local temperature, particularly when strong mitigation is employed later in the simulation. Delta method patterns were more consistent across scenarios with less heterogeneity in local temporal and spatial GMT sensitivity. With the assumption that different future forcing scenarios should not change the resulting pattern, the delta pattern is more consistent across scenarios, regardless of epoch chosen, despite differences in epoch trends being large.

Our pattern library was created because the online tools and software that generates pattern scaling products do not provide pattern data and diagnostics, and do not offer flexibility in use of a SCM for scaling. We have created a library of patterns with descriptive statistics for each output file, which we believe to be beneficial for uncertainty quantification and probabilistic statistical analysis.

Creation of a pattern library is the first step in our goal of exploring inter-model and future forcing uncertainty in climate projections. Our next steps will be to push the current boundaries of pattern scaling by exploring sub-annual pattern scaling, scaling measures of climate variability, and scaling of different variables, such as pH. Our efforts will be documented in future manuscripts, and all patterns will be added to the repository.

## 5 Pattern Library

The pattern library is available on GitHub through the Joint Global Change Research Institution repository (https://github.com/JGCRI/). The purpose of creating this pattern library was to allow for researchers across various fields to be able to efficiently use the statistical patterns generated by the described regression method to examine model response to change in global mean temperature for all the available CMIP5 models (41 models, at present). We also further intend for those patterns to be easy to scale using a scaler generated from a SCM of ones choosing. To this end, included in each netCDF file for each model is:

1. The individual model pattern (2-D);





2. The adjusted $R^2$ between the predictor and dependent terms (2-D);

3. The regression error term (2-D), which can be used to construct a residual time series;

4. A historical climatology based on the 1960-1999 average from each model (2-D), which can be used to construct absolute values at time $X$;

5. The $95^{th}$ percentile confidence level pattern for model parameters.

The patterns range in size (1 MB to 165 KB) due to spatial resolution, but all patterns were kept at the native resolution of the dependent variable. This was done to retain model specific information, which may have been lost if regridded to a common spatial resolution.

All source code used to produce patterns is available in the aforementioned repository. Source code is written in NCAR Command Language (Version 6.3.0; http://dx.doi.org/10.5065/D6WD3XH5).

## 6   Code and/or data availability

CMIP5 model data is publicly available via the Earth System Grid Federation website (ESGF, https://pcmdi.llnl.gov/). Code used to construct this analysis is available on GitHub through the Joint Global Change Research Institution repository

(https://github.com/JGCRI/CMIP5_patterns/). Any additional data can be obtained from Cary Lynch (cary.lynch@pnnl.gov).

*Acknowledgements.*  This research is based on work supported by the US Department of Energy, Office of Science, Integrated Assessment Research Program. The Pacific Northwest National Laboratory is operated for DOE by Battelle Memorial Institute under contract DE-AC05-76RL01830.





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



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



**Table 1.** List of the CMIP5 models and their respective spatial resolution and organization used in this analysis.

| Models | Spatial Resolution | Organization |
|---|---|---|
| ACCESS1-0 | 1.2 x 1.8 | Commonwealth Scientific and Industrial Research Organization, and Bureau of Meteorology, Australia |
| CanESM2 | 2.8 x 2.8 | Canadian Centre for Climate Modeling and Analysis, Canada |
| CCSM4 | 1 x 1 | NCAR, University Corporation for Atmospheric Research, U.S.A |
| CMCC-CMS | 1.8 x 1.8 | Centro Euro-Mediterraneo per Cambiamenti Climatici, Bologna, Italy |
| CNRM-CM5 | 1.4 x 1.4 | Centre National de Recherches Meteorologiques/Center Europeen de Recherche et Formation Avancees en Calcul Scientifique, France |
| CSIRO-Mk3.6 | 1.8 x 1.8 | Commonwealth Scientific and Industrial Research Organization, Australia |
| GFDL-CM3 | 2 x 2 | NOAA, Geophysical Fluid Dynamic Laboratory, U.S.A |
| HadGEM2-ES | 1.2 x 1.8 | Meteorological Office Hadley Centre, UK |
| INMCM4 | 1.5 x 2 | Institute for Numerical Mathematics, Russian Academy of Sciences, Russia |
| IPSL-CM5A-MR | 1.25 x 2.5 | Laboratoire de Meteorologique Dynamique, Institut Pierre-Simon Laplace, France |
| MIROC-ESM | 3 x 3 | Atmosphere and Ocean Research Institute, National Institute for Environmental Studies, and Japan Agency for Marine-Earth Science and Technology, Japan |
| MPI-ESM-MR | 1.8 x 1.8 | Max Planck Institute for Meteorology, Germany |
| NorESM1-M | 2 x 2 | Norwegian Climate Centre, Norway |

**Table 2.** Root mean square error between actual and pattern predicted anomalies in $^{o}$C/ $^{o}$C for each pattern methodology.

| | | L21C/L19C | L21C/L20C | LSR |
|---|---|---|---|---|
| **Annual** | **rcp8.5** | 1.562 | 1.104 | 0.388 |
| | **rcp4.5** | 1.491 | 1.031 | 0.341 |
| **DJF** | **rcp8.5** | 1.786 | 1.267 | 0.586 |
| | **rcp4.5** | 1.727 | 1.204 | 0.496 |
| **JJA** | **rcp8.5** | 1.361 | 0.950 | 0.343 |
| | **rcp4.5** | 1.301 | 0.889 | 0.282 |

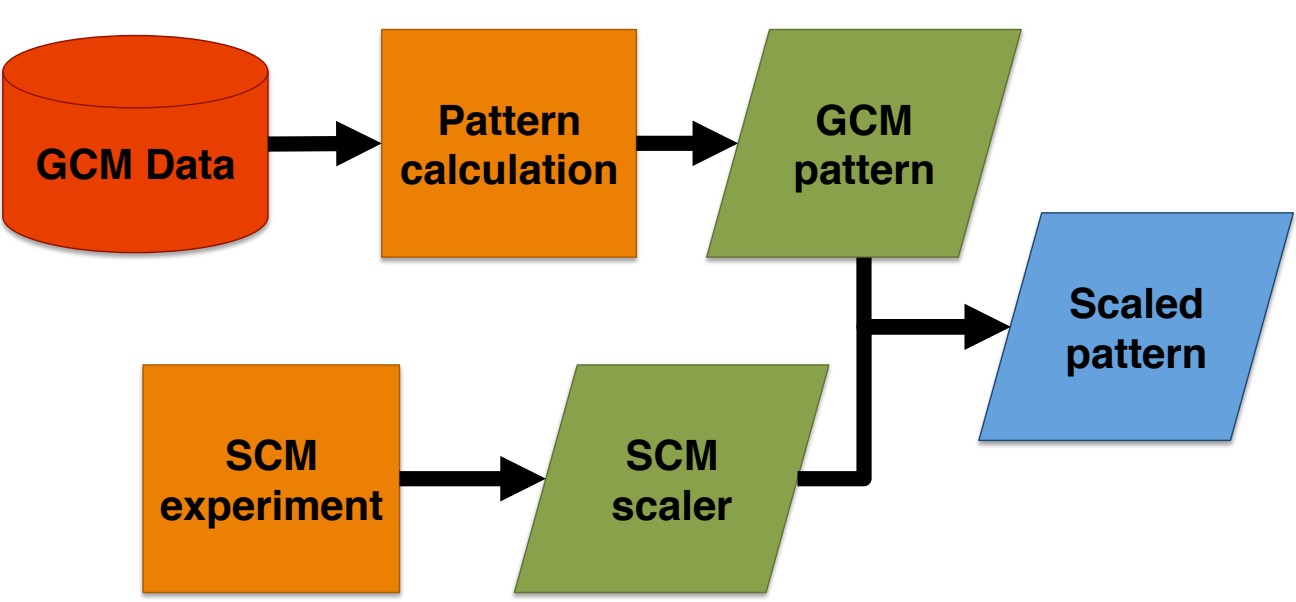

**Figure 1.** Generalized flowchart of pattern scaling process.





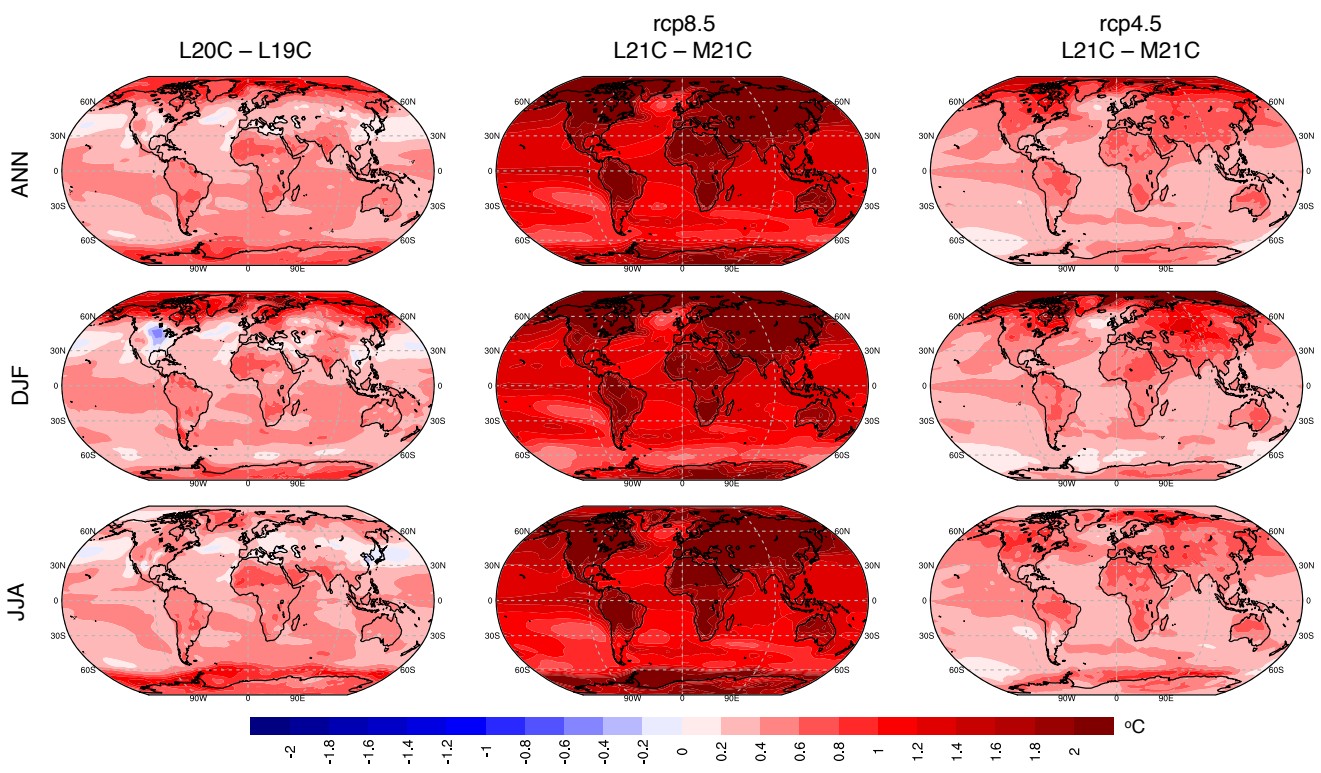

**Figure 2.** Ensemble mean temperature difference (°C) between L20C (1971-2000) and L19C (1861-1890) epoch, L21C (2071-2100) and M21C (2041-2070) epoch from rcp8.5 scenario, and L21C and M21C epoch from rcp4.5 scenario for mean annual, DJF, and JJA.





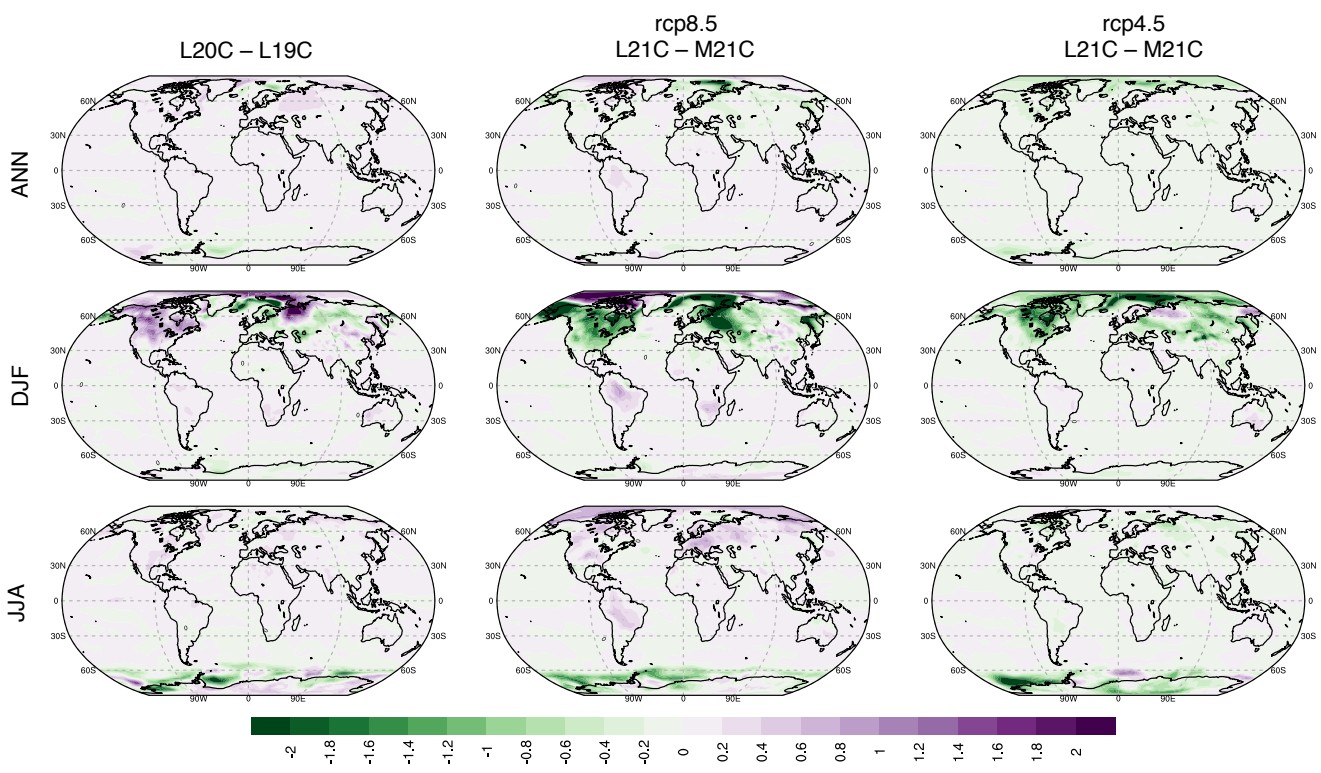

**Figure 3.** Ensemble mean temperature variance difference between L20C (1971-2000) and L19C (1861-1890) epoch, L21C (2071-2100) and M21C (2041-2070) epoch from rcp8.5 scenario, and L21C and M21C epoch from rcp4.5 scenario for mean annual, DJF, and JJA.




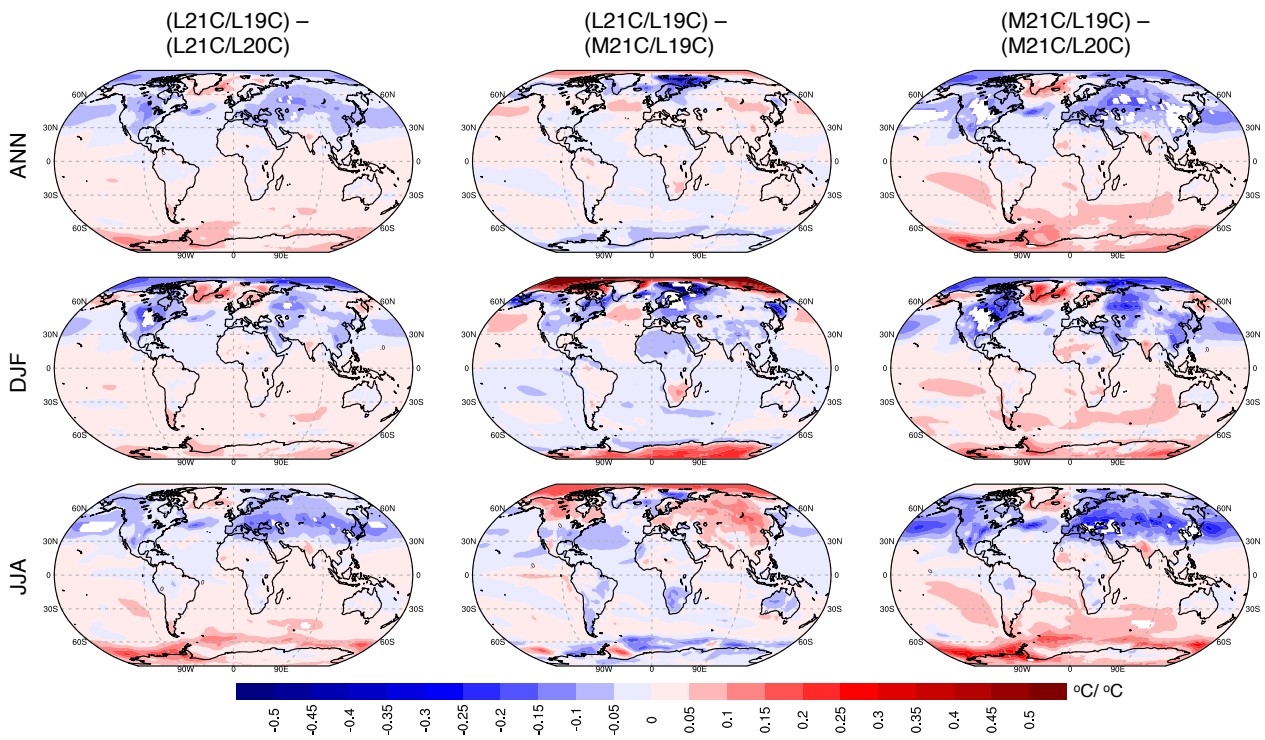

**Figure 4.** Ensemble mean delta method pattern differences between L21C (2071-2100)/ L19C (1861-1890) and L21C/ L20C (1971-2000); L21C/ L19C and M21C (2041-2070) / L19C; and M21C/ L19C and M21C/ L20C for mean annual, DJF, and JJA. Significance values below the 95% confidence interval using a Student's t-distribution probability statistic were masked.





**Figure 5.** Ensemble mean regression method pattern and delta method pattern differences in $^{o}C/\,^{o}C$ for L21C (2071-2100)/ L19C (1861-1890) and L21C/ L20C (1971-2000) for annual, DJF and JJA for future forcing scenario rcp8.5. Significance values below the 95% confidence interval using a Student's t-distribution probability statistic were masked.

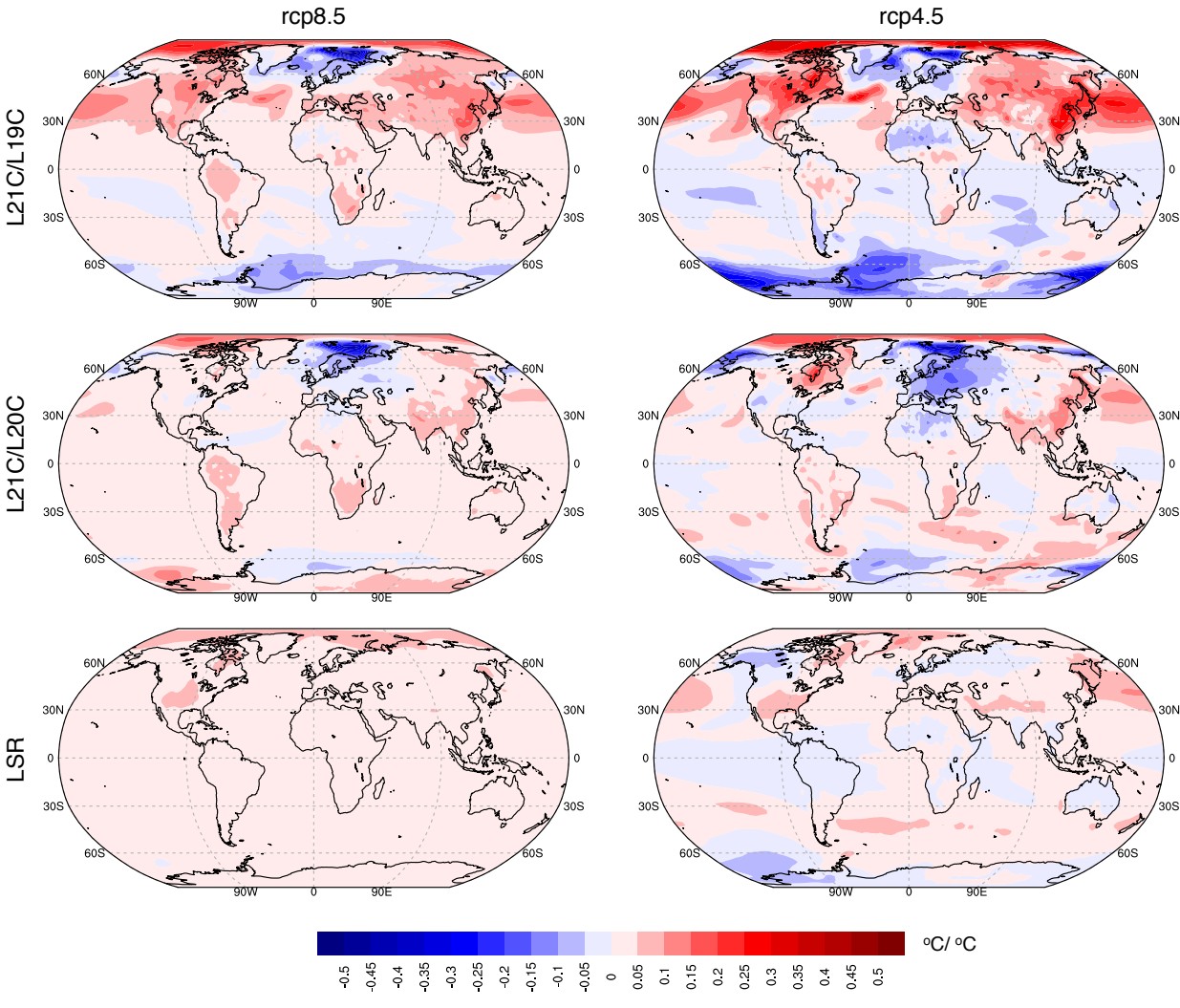

**Figure 6.** Difference between annual modeled change and predicted pattern, when GMT change = 1 for future forcing scenarios rcp8.5 and rcp4.5 in $^{o}$C/ $^{o}$C. Differences were shown between ensemble mean modeled GMT change and L21C (2071-2100)/ L19C (1861-1890), L21C/ L20C (1971-2000), and LSR patterns.





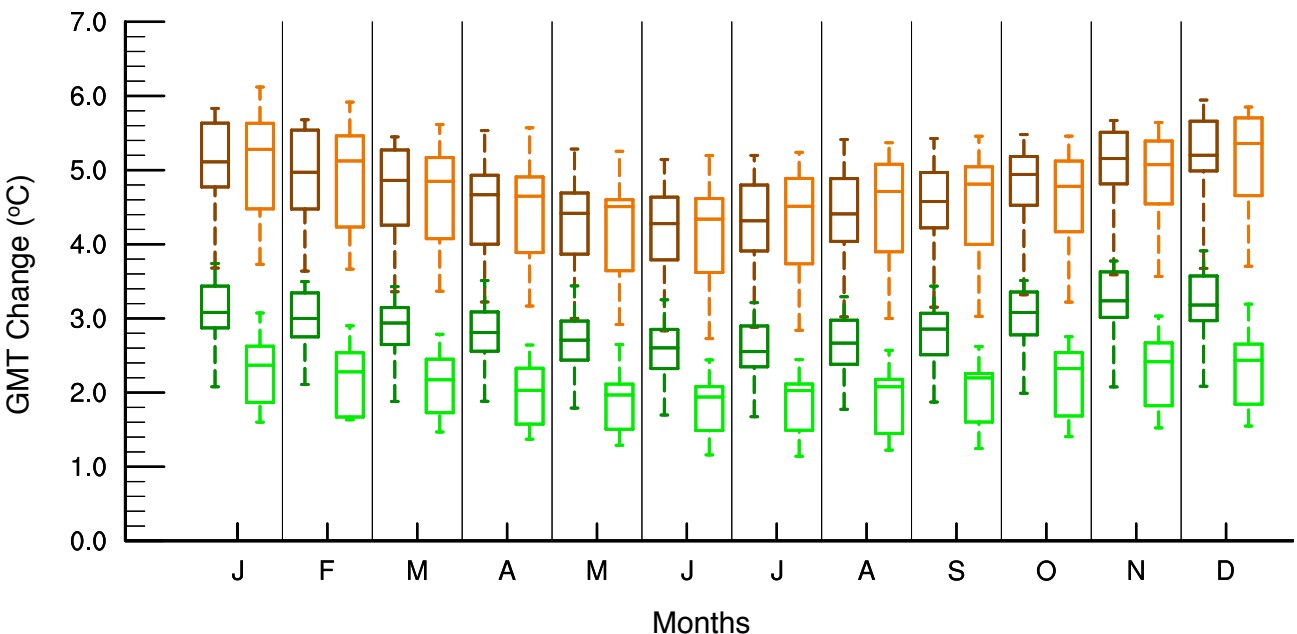

**Figure 7.** Ensemble spread in GMT change ($^o$C) for L21C (2071-2100)/ L19C (1861-1890) epoch (brown (rcp8.5) and darkgreen (rcp4.5)) and 21$^{st}$ Century (2006-2100) trend (orange (rcp8.5) and light green (rcp4.5)).





**Figure 8.** Ensemble annual average difference in $^{o}$C/ $^{o}$C and significance of difference using a Student's t-distribution probability statistic between future forcing scenarios rcp8.5 and rcp4.5 for L21C (2071-2100)/ L19C (1861-1890), L21C/ L20C (1971-2000), and LSR patterns.





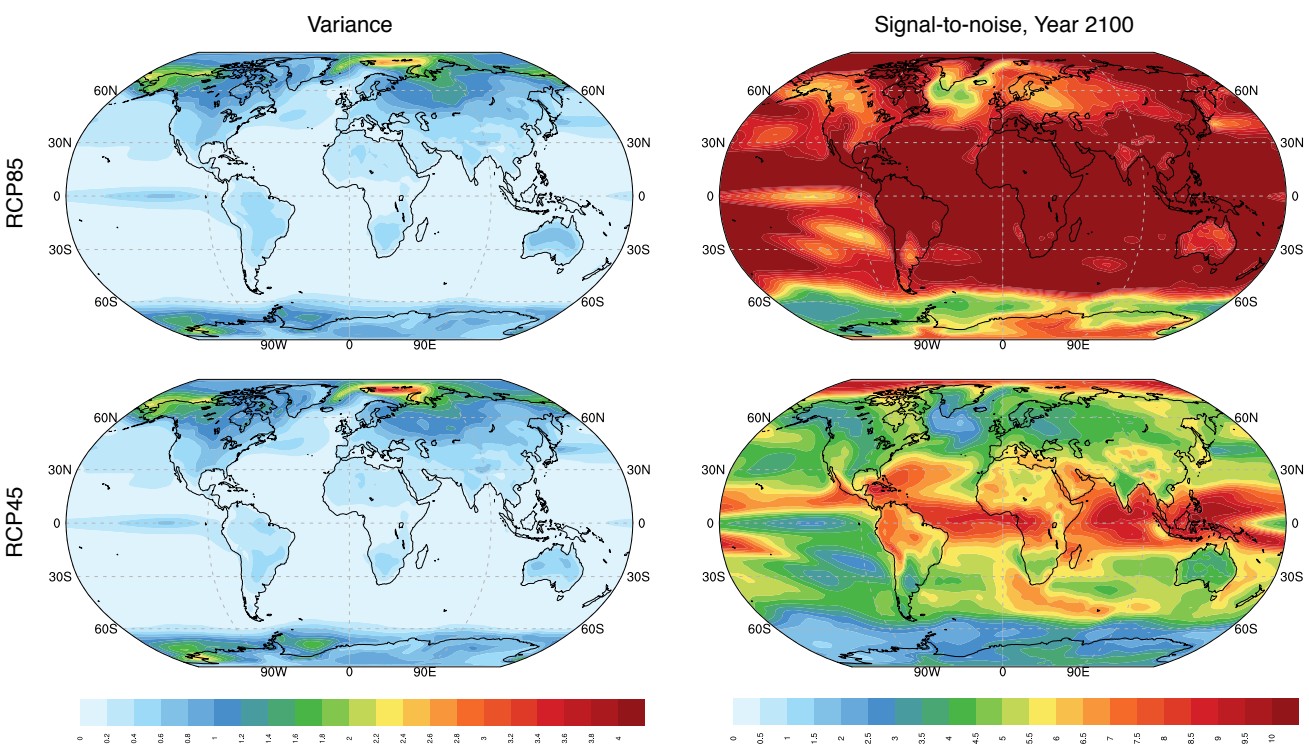

**Figure 9.** Ensemble mean detrended variance for 21$^{st}$ Century and signal to noise ratio at year 2100 for rcp8.5 and rcp4.5.



**Figure 10.** Ensemble mean R-squared of LSR patterns for rcp8.5 and rcp4.5 scenarios for annual mean, DJF, and JJA.





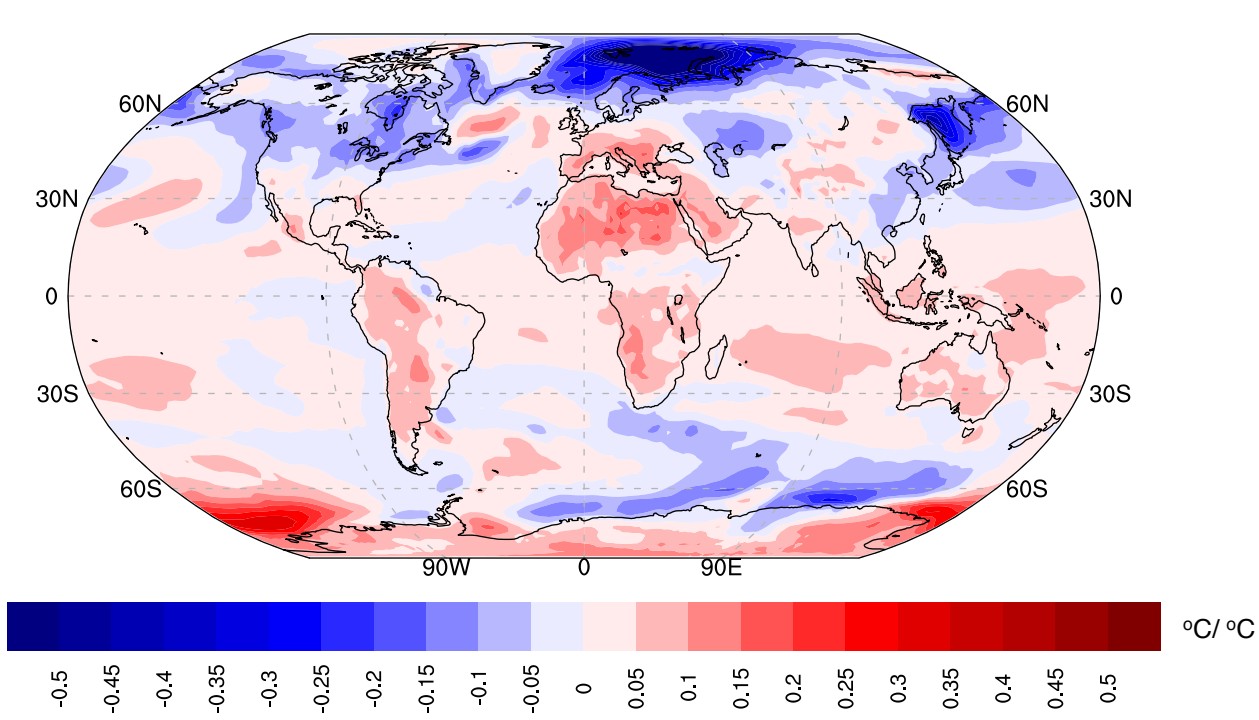

**Figure 11.** Ensemble annual average difference in $^{o}$C/ $^{o}$C in the ratio of local/global linear trend over the $21^{st}$ Century (2006-2100) between rcp8.5 and rcp4.5.



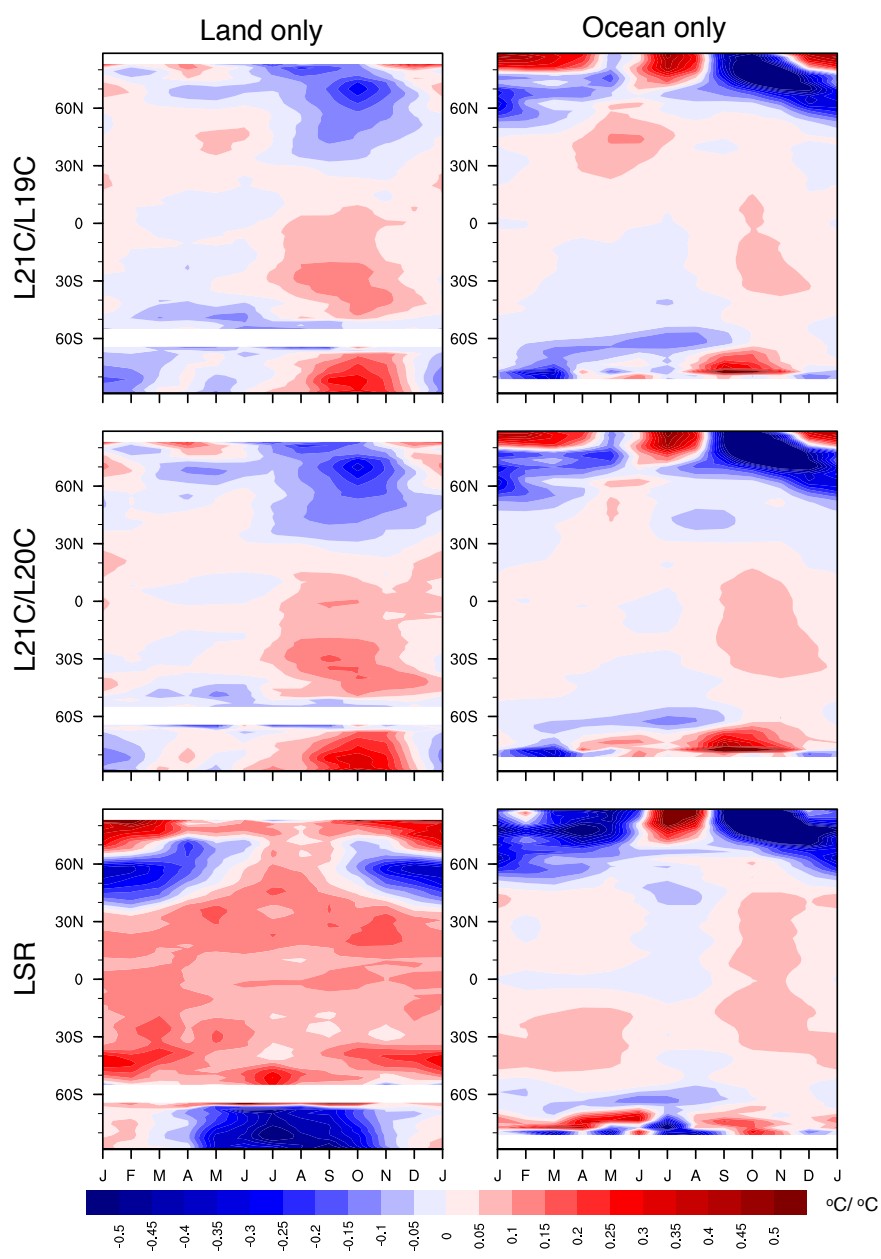

**Figure 12.** Ensemble mean temperature pattern differences in $^{o}$C/ $^{o}$C between rcp8.5 and rcp4.5 in zonal monthly means for land only and ocean only for L21C (2071-2100)/ L19C (1861-1890), L21C/ L20C (1971-2000), and LSR patterns.