# Peer review of "An open-access CMIP5 pattern library for temperature and precipitation: Description and methodology."

_Earth System Science Data, 2016_

## Referee Comment (RC1) · Anonymous Referee #1 · 3 Mar 2017

Manuscript reviewed: "An open-access CMIP5 pattern library for temperature and precipitation: Description and methodology"

Authors: Cary Lynch, Corinne Hartin, Ben Bond-Lamberty, Ben Kravitz

Journal: Earth System Science Data

Date: 3 March 2017

The authors have constructed a dataset that includes scaling patterns for a wide number of global climate models (GCMs). These patterns can be utilized in creating future projections for such forcing scenarios that have not been explicitly simulated by the models.

Recommendation: The dataset is accessible but the manuscript needs substantial revisions.

**1 Data evaluation**

- Uniqueness: rating = 2

- Usefulness: rating = 1

- Completeness: rating =1

- Data quality: rating = 2

- Comments: The methods used for creating the data (linear regression etc.) are not new but the effort has been substantial and has required much resources. Scaling patterns are available for 41 GCMs at monthly, seasonal and annual level. For controlling purpose, I downloaded one file; the file was easily accessible. This example file was readable by all three softwares/commands that I proved: grads, ncdump and cdo.

  One problem is that the files must be downloaded one by one. Regarding the large number of files, a wget tool similar to that in use in the CMIP5 data bank would be useful for those users that are interested in a large ensemble of models. Such a tool might be developed later, and this issue is not any reason to delay the publication of the database and article.

  It is impossible for me to check the correctness of the data included in the files; we have to trust that the team has done their work properly in this respect.

**2 Manuscript: Main comments**

The authors have used much space to show that the regression method outperforms the delta-change method in producing scaling patterns. However, these details do not constitute necessary information for the users of the database. For example, exploring the dependence of delta change patters on the reference and scenario periods is quite far from the main focus of the paper. (If this discussion is shortened, some more detailed comments presented below may become irrelevant.)

There is one very good additional reason to provide scaling patters just for the regression method and not for the delta-change method: the delta-change patterns are much more straightforward to calculate by the users themselves than the patterns based on the regression method. This might be mentioned in the paper as a motivation for selecting this method.

I find that sections 1–2 are moderately good but the authors should have edited section 3 much more carefully.

1. Page 1, line 6-7: "This paper presents patterns from all CMIP5 models for temperature and precipitation...". In fact, the paper only evaluates the temperature patterns but no results are given for precipitation. Even so,

precipitation files are available in the repository as well. Accordingly, a brief evaluation of the precipitation patterns should indeed be included in the paper (pattern minus original model output).

2. In the repository, patterns are given for 41 GCMs, but in the manuscript, for some reason, the evaluation of the methodology and patterns is limited to 12 GCMs only. No motivation for this choice is given. Studying a wider ensemble of GCMs would give a more robust picture on the differences between the methods, scenarios, etc. In particular, it is strange that no GCM from China (e.g., BCC-CSM1-1) has been included in the ensemble!

3. As a criterion for the similarity of the multi-model mean patterns, you use the Student's t test (page 4, line 30 onwards). As the sample is small (12 GCMs), the test only reveals very large differences between the means. Accordingly, the t value remaining below the significance threshold does not provide any strong evidence for the actual similarity of the means. Text (on page 5, lines 23–24 and 29; page 6, lines 11–12,...) would also need revision.

   Perhaps, it is not so essential to study in detail whether the differences in the patterns are statistically significant. It is trivial that the pattern depends, to some extent, on the RCP scenario, reference/scenario period, pattern-scaling method, etc. Whether these differences are statistically significant or not, largely depends on the size of ensemble. The larger the number of GCMs, the smaller differences would be significantly different. You have selected to provide patterns founded on the linear regression, and, in my opinion, any inter-method differences do not invalidate that choice.

4. I do not find the comparison presented in the first paragraph of section 3.2 relevant. The outcome is fairly trivial: if one compares dissimilar periods, the GMT change is different as well. The similarity of the changes under the RCP8.5 scenario appears fortuitous. In particular, the values presented in Fig. 7 do not give the rate of GMT change, since the length of period in L21C/L19C and the 21st century is far from the same.

5. Caption of Fig. 6 is unclear. I presume that "modelled change" refers to the multi-model mean change derived directly from the original model output while "predicted pattern" stands for the pattern calculated by one of the pattern-scaling methods. This should be stated more clearly. Moreover, if one wants to elucidate the performance of the scaling methods, it would be more logical to show the difference (scaled value) minus (reference, i.e., the original model output). In the present representation, positive values evidently refer to an underestimation (page 6, line 14), which makes the interpretation difficult for the reader. Finally, I did not understand "when GMT change = 1 (without any unit!)"; perhaps the idea is that the maps have been normalized to correspond to an 1°C increase in GMT. In that case, do you have any explanation why the global mean of the difference is not equal to zero?

6. In Table 2, there is something that I do not understand. In Fig. 6, the differences are far $< 1$ everywhere. How can you obtain RMS differences that are much larger than the differences occurring at any individual grid point? Regarding these large RMS differences, how can you state that "Nevertheless, Table 2 indicates that the both methodologies do well emulating actual model output."

**3 Specific comments**

1. Page 1, lines 10 and 14: The magnitude of the actual temperature increase depends on the radiative forcing. It would be relevant to give here normalized responses in °C/°C rather than the absolute responses.

2. In the www address given in the abstract (http://doi.org/10.5281/zenodo.235905), I only found the annual mean patterns. By contrast, in the address stated in section 6 (https://github.com/JGCRI/) there was a full manifold of files.

3. In the abstract, it should be explicitly stated that the patterns in the repository are based on the least square regression method rather than on the delta-change method.

4. Page 4, line 27: In order to reproduce the time series of $TL_{MS}$, $\varepsilon$ should be three- rather than two-dimensional (a function of time in addition to the latitude and longitude).

5. On page 6, there are sentences that are hard to understand, e.g.: "To evaluate performance of each pattern methodology, accuracy was based on how well the patterns approximated the linear GMT change of 1°C simulated by each GCM."

6. Page 6, line 18–19: "Overall, it appears that the regression pattern scaling method underestimates the relationship between global temperature and local temperature, but the degree to which it overestimates the relationship is small ($< 0.08$°C)." Some interpretation should be given for this. Is this difference in global mean temperature change induced by the use of a simple climate model?

7. Page 7, line 12–13: The nonlinear evolution of temperature is related to the retreat of sea ice.

8. Page 7, line 16: $R^2$ evidently refers to the square of correlation coefficient; this should be mentioned explicitly. Is the correlation calculated with respect to time?

9. Page 7, line 20: "despite lower local and global trends". This might be removed. The ratio may well be large regardless of the magnitude of the numerator and denominator.

10. Page 7, line 21: Over oceans in winter, the rate of warming is large before the ice melt. After the melt, the large thermal inertia of the sea water tends to decelerate warming.

11. Page 9, line 3: I studied one file by the ncdump command, and in that file the period for historical climatology was 1961–1990.

12. Page 8, lines 13–14: "particularly when strong mitigation is employed later in the simulation". The idea is not clear for me.

13. In the Figures, colour hues are very close to one another, and thus it is hard to infer the actual values of the field at some specified location. For the largest values, it would be good to use some other colours than red and blue. Alternatively, you might add some contours to facilitate interpretation.

14. In Figs. 4 and 6, some other colour than white should be used to emphasize areas with statistically significant differences.

15. Fig. 3: Is the variance calculated with respect to time? Standard deviation might be more illustrative than the variance. Moreover, the variability of temperature is very different in the different parts of the world. Therefore, it would be relevant to give differences in std in a relative sense (e.g., $(\sigma_1 - \sigma_2)/\sigma_2$). This would reveal where the differences actually are pronounced. After seeing the differences in rms in relative terms, it is possible to evaluate the statement "Differences in variance across epochs were also small" (page 5, line 19) and whether "these relatively small differences in variance (or std?) between epochs were not likely to affect the resulting temperature patterns".

**4   Minor comments**

1. Page 2, line 13: "adaptability of additional predictors" is difficult to understand.

2. Page 2, line 20: Does "... vary between scenarios and for other climate variables...." mean "... vary between scenarios and climate variables"?

3. Page 2, line 31–33: "In linear regression, only the error term is assumed to have a normal distribution, so it is likely that climate extremes would yield high error terms." Please explain this more explicitly.

4. Page 4, line 12: "a 30-year reference epoch of 1986–2005". Only 20 years!

5. Page 4, line 15–17: The sentence should be rephrased.

6. Page 4, line 18: 1861–1990 should be 1861–1890?

7. Page 5, line 2: "ensemble variance". Variance within or between the ensembles? Please specify.

8. Page 5, line 22: "in the observed period". You are dealing with model simulations rather than observations.

9. Page 7, line 6: Should the unit be °C/°C rather than °C?

10. Page 8, line 6: I did not understand "The local to global fit is strong".

11. Page 8, line 8: Give some examples of potential additional predictors.

12. Page 8, line 16: "epoch trends"?

13. Page 8, line 26: Please define GitHub.

14. Page 9, line 6: 165 kB to 1 MB.

15. In the pdf file of the manuscript, list of references is given twice.

16. In Table 2, are the rms differences global?

17. In the caption of Fig. 4, specify the RCP scenario.

18. Fig. 7: The unit is different for change (°C) and trend (°C/year).

19. In Fig. 9, consider showing the standard deviation rather than variance; give the unit of the quantity in the caption.

20. Fig. 10, caption: Replace "R-squared" by "the square of the correlation coefficient".

21. Fig. 11: Is this "Ensemble mean annual average"?

---

## Referee Comment (RC2) · Anonymous Referee #2 · 12 Mar 2017

Summary: The paper presents a library of patterns from CMIP5 models, generated using two published methodologies, which are then compared. The value of the work lies in the fact that it takes a lot of time and resources to compute these spatial patterns, so making it available to the community would facilitate advances in this field. Also, a standardized comparison helps users to recognize the strengths and weaknesses of the two methodologies. The results are provided to the public, packaged with documentation, code, etc., and associated with a doi.

This paper appears to be written out as a science report with a data, methods and results section, rather than a dataset description. I expected an explanation of the library content, i.e., structure of files, as well as how to access, and applications. The

paper is well written and clear, but it is the content and organization that could be improved. I suggest a re-write for greater clarity to potential users of this dataset. In particular, Section 5 seems to be added at the last minute. This is the section that users will read for a description of the data so it could be expanded, and information in other sections could be moved here. As such, it could be inserted earlier. Also, the two access points of the library do not have the same contents, and are not fully described in the article reviewed. In fact, my impression from reading the paper was that results for both methodologies were available, and that it was limited to 12 models. It took several re-reads, and examination of the files to understand that this was not the case.

Specific comments.

1. I had was confused by the description provided in that it discusses the application of methodology to only 12 models that met certain criteria needed for the comparison, and only the first realization. It is not clear what was done for these other models for which there are files are available at the access sites. Does the library contain results for both methods? The netCDF file contained only temperature and precipitation output, seemingly for one method only.

2. The paper shows results for temperature, but the library contains precipitation. Since these two variables behave quite differently on a global and temporal scale, it would be worth showing some precipitation examples, and perhaps go less in depth about temperature.

3. There should be a clearer explanation about the difference between the two access points. At the doi location (mentioned in the abstract), one can download the entire packaged zipped. Once unzipped, the README is hard to open because it contains scripting language so the laptop software refuses to open it. Only annual patterns are found here. In the github location (mentioned in the section 5), one can download only single files, but the README displays easily. Once downloaded, NetCDF files from both locations open easily. Also, one site appears to have only annual files.
4. Rather than merely listing the variables, it would have been more useful to a potential user to get a brief recap of what the variables represent or refer back to the appropriate section. For example, the grid size depends on the original model grid. Why not refer back to Table 1 for the 12 models? What about the other models? Also it was not clear if the patterns were derived on the coarse grid and then remapped, because the comparison was done on a common grid. Where is the description of the analysis done on native grid?

5. I cannot validate this product. The closest thing to a "validation" for this type of product might be a computation of the scaling patterns for the last 20-50 years, using the decades prior to that as the baseline, i.e., an L20C/M20C. In this way, the user can see how a model captured the changes in the last few decades, and compare with actual observed trends. This might help them users decide which model(s) might be more appropriate for this use. If this work has been done by others, a brief review of that literature would be very helpful and should be included in the paper. In re-reading this paper, it appears that the Data and results might actually represent the Validation of the dataset.

6. I would also suggest a polar projected example (Arctic) for temperature, at least.

7. The dataset is not complete in the sense that it only analyzes the for each model, it only analyzed one run, as stated in the paper. Thus, the library has limited value than if they analyzed all runs. Not all users will want the results for only one run, so they will have to re-compute the patterns anyway.

8. I cannot determine if the dataset is of high quality that but it is reassuring that error fields and statistics are provided.

9. I pasted the link listed under associated files listed for the climatology variable. It did not go anywhere. I am surprised SI units are not used. Perhaps there is a specific reason for this? It should be explained in the manuscript.

10. I didn't do a detailed list of the typos, which are few. However, the paper will need to be re-organized.

Minor comments: Page. 4 1861 -1990 ...should be 1861-1890?

Introduction: Please clarify if the patterns are purely spatial, not temporal.

Table 1. What are the units? Is it longitude X latitude or vice versa?

Table 2. Need more clarity on what this represents..difference in RMS?

[Figure]

---

## Editor Comment (EC1) · M. E. Contadakis (Editor) · 14 Mar 2017

I am much obliged to both reviewers for their meticulus review. Both reviewers find the paper interesting and very usefull to active scientist in the field. Off course they make their comments and recomentations which I believe are justified. I hope that these reviews will stimulated further discution not only by the authors but by other interesting sientists.

---

## Editor Comment (EC2) · M. E. Contadakis (Editor) · 23 Mar 2017

Dear Aythors You are encouraged to publish a final response to all comments by 17 Apr 2017 best regards M.E.Contadakis Topical Editor

---

## Author Comment (AC1) · 12 Apr 2017

Manuscript: "An open-access CMIP5 pattern library for temperature and precipitation: Description and methodology"

Authors: Cary Lynch, Corinne Hartin, Ben Bond-Lamberty, Ben Kravitz

Journal: Earth System Science Data

Date: 12 April 2017

Response to Reviewer #1

**Data Evaluation Comments:**

Comment: "One problem is that the files must be downloaded one by one. Regarding the large number of files, a wget tool similar to that in use in the CMIP5 data bank would be useful…"

Response: The benefits of a repository on GitHub is that you can download and/or clone the entire repository in a short amount of time. Or one can choose a specific model to download. We agree that a wget script would be very useful, and we would be interested in adding this feature in the future.

**Manuscript: Main Comments**

Comment: The authors have used much space to show that the regression method outperforms the delta-change method in producing scaling patterns. However, these details do not constitute necessary information for the users of the database. For example, exploring the dependence of delta change patters on the reference and scenario periods is quite far from the main focus of the paper.

Response: The SCENGEN (Wigley et al, 2008) and the CLIMGEN (Osborn et al, 2009) pattern scaling repositories use the delta method to construct their patterns. As such, we wanted to provide justification as to why users may want to consider our repository. In the process, we have provided some interesting analyses that provide novel scientific value regarding differences between simple emulation techniques. That said, we agree that the focus of the manuscript should not be on epoch choice. To this end we have shortened our discussion of epoch differences and removed Figures 3 & 7.

Comment: Page 1, line 6-7: "This paper presents patterns from all CMIP5 models for temperature and precipitation...". In fact, the paper only evaluates the temperature patterns but no results are given for precipitation. Even so, precipitation files are available in the repository as well. Accordingly, a brief evaluation of the precipitation patterns should indeed be included in the paper (pattern minus original model output).

**Response:** Because the relationship between global mean temperature and local precipitation is complex and requires a more in-depth examination of pattern scaling methodologies, we have written an entirely separate manuscript discussing precipitation pattern scaling (Kravitz et al., 2017). We reference that paper in our manuscript and have revised some of our description to make clear why the repository contains precipitation patterns.

**Revision:** This paper presents a pattern library with all available CMIP5 models for temperature and precipitation on an annual and sub-annual basis, along with the code used to generate these patterns.

**Comment:** In the repository, patterns are given for 41 GCMs, but in the manuscript, for some reason, the evaluation of the methodology and patterns is limited to 12 GCMs only. No motivation for this choice is given. Studying a wider ensemble of GCMs would give a more robust picture on the differences between the methods, scenarios, etc. In particular, it is strange that no GCM from China (e.g., BCC-CSM1-1) has been included in the ensemble!

**Response:** This is a good point. For this analysis we did not want to use a large ensemble of models, and we also did not want to arbitrarily choose a multi-model ensemble that would over-represent particular sub-models (Sanderson et al, 2015). For these reasons, we developed a small set of simple performance metrics for a 10-12 model ensemble that realistically produced mean observed spatial/temporal climate. We then used a representative model from each modeling center, to limit the number of models used in this analysis. The few metrics used served primarily to exclude models that were statistically different from the observed data, in this case NCEP-NCAR reanalysis (Kalnay et al, 1996). We stress that accurate simulations of global mean temperature do not mean that a model is 'better' or that 'good' models will have better predictive skill. We have added some explanation to the supplementary material, along with metrics of model performance according to the particular attributes in which we were interested

The issue of model independence is important. The bcc-csm1-1 and bcc-csm1-1-m were not included in this analysis because their atmospheric and land models are largely based on the NCAR CAM models, and their ocean and sea-ice models are largely based on the GFDL ocean and sea ice models. As such, we deemed them to be too similar to models that were already included in our ensemble, even though both bcc models performed well in the simple metrics evaluation. The calculated patterns for both models are available in the data repository.

**Comment:** As a criterion for the similarity of the multi-model mean patterns, you use the Student's t test (page 4, line 30 onwards). As the sample is small (12 GCMs), the test only reveals very large differences between the means. Accordingly, the t value remaining below the significance threshold does not provide any strong evidence for the actual similarity of the means. Text (on page 5, lines 23–24 and 29; page 6, lines 11–12,...) would also need revision.

Perhaps, it is not so essential to study in detail whether the differences in the patterns are statistically significant. It is trivial that the pattern depends, to some extent, on the RCP scenario, reference/scenario period, pattern- scaling method, etc. Whether these differences are statistically significant or not, largely depends on the size of ensemble. The larger the number of GCMs, the smaller differences would be significantly different. You have selected to provide patterns founded on the linear regression, and, in my opinion, any inter-method differences do not invalidate that choice.

Response:  Agreed, our interpretation of the results from the Student's t test is limited due to the small ensemble size.  However, historically, the first demonstration of the t-test was in an application to sample sizes of size *4*.  Obtaining improved results (over a z-test, for example) for *small* samples is a t-test's asset (york.ac.uk/depts/maths/histstat/student.pdf).

Overall, the difference in patterns generated from each methodology between the 41 member ensemble and the 12 model ensemble is small, except for JJA at high latitudes. However, the areas where there are significant differences, based on the Students ttest statistic are different (see below figure).  For the 41 member ensemble there are more regions of significance differences in the mid-latitudes.  This may be the result of a few models that skew the ensemble mean or variance.

[Figure]

*Figure 5.*  Ensemble mean regression method pattern and delta method pattern differences in ºC/ºC for L21C (2071-2100)/ L19C (1861-1890) and L21C/ L20C (1971-2000) for annual, DJF and JJA for future forcing scenario rcp8.5.  Significance values below the 95% confidence interval using a Student's t-distribution probability statistic were masked.

We agree that the differences in patterns with respect to epoch and scenario are minor, which is precisely the point; the patterns are largely the same regardless of method, scenario, epoch, and ensemble. This gives confidence in the use of patterns in simple climate model experiments, regardless of methodology.

Comment: I do not find the comparison presented in the first paragraph of section 3.2 relevant. The outcome is fairly trivial: if one compares dissimilar periods, the GMT change is different as well. The similarity of the changes under the RCP8.5 scenario appears fortuitous. In particular, the values presented in Fig. 7 do not give the rate of GMT change, since the length of period in L21C/L19C and the 21st century is far from the same.

Response: After further review, we agree that this figure is not germane to the discussion of pattern differences across scenarios for the above stated reasons. This figure and the corresponding text have been removed.

Comment: Caption of Fig. 6 is unclear. I presume that "modelled change" refers to the multi-model mean change derived directly from the original model output while "predicted pattern" stands for the pattern calculated by one of the pattern-scaling methods. This should be stated more clearly. Moreover, if one wants to elucidate the performance of the scaling methods, it would be more logical to show the difference (scaled value) minus (reference, i.e., the original model output). In the present representation, positive values evidently refer to an underestimation (page 6, line 14), which makes the interpretation difficult for the reader. Finally, I did not understand "when GMT change = 1 (without any unit!)"; perhaps the idea is that the maps have been normalized to correspond to an 1∘C increase in GMT. In that case, do you have any explanation why the global mean of the difference is not equal to zero?

Response: Yes, thank you, this caption was not clearly worded. It has been edited for clarity. There was no normalization, and a unit for GMT is needed. This has been corrected in the caption as well.

We agree that it would be more logical to show the difference between the scaled pattern and the model mean change, rather than the reverse. This figure has been redone to reflect this.

Comment: In Table 2, there is something that I do not understand. In Fig. 6, the differences are far < 1 everywhere. How can you obtain RMS differences that are much larger than the differences occurring at any individual grid point? Regarding these large RMS differences, how can you state that "Nevertheless, Table 2 indicates that the both methodologies do well emulating actual model output."

Response: Thank you for pointing that out. We misplaced a decimal point and have corrected the table in the updated manuscript.

**Specific Comments:**

Comment: Page 1, lines 10 and 14: The magnitude of the actual temperature increase depends on the radiative forcing. It would be relevant to give here normalized responses in °C/°C rather than the absolute responses.

Response: We agree that a normalized response would be more relevant, however we are limiting the discussion of epoch differences and this line in the abstract has been taken out.

Comment: In the abstract, it should be explicitly stated that the patterns in the repository are based on the least square regression method rather than on the delta-change method.

Response: Yes, thank you. The abstract has been edited to make this clear.

Comment: Page 4, line 27: In order to reproduce the time series of $T\,LM\,S$ , $\varepsilon$ should be three- rather than two-dimensional (a function of time in addition to the latitude and longitude).

Response: Yes, thank you. This line has been edited.

Comment: On page 6, there are sentences that are hard to understand, e.g.: "To evaluate performance of each pattern methodology, accuracy was based on how well the patterns approximated the linear GMT change of 1°C simulated by each GCM."

Response: Agreed. Page 6, lines 13-15, have been edited.

Revision: To evaluate pattern performance from each methodology, we compared the generated patterns to the ensemble local mean change when the linear GMT change is 1°C. The delta patterns largely underestimate the spatial pattern, particularly over land and mid-high Northern latitudes (Figure 6).

Comment: Page 6, line 18–19: "Overall, it appears that the regression pattern scaling method underestimates the relationship between global temperature and local temperature, but the degree to which it overestimates the relationship is small (< 0.08°C)." Some interpretation should be given for this. Is this difference in global mean temperature change induced by the use of a simple climate model?

Response: No, a scaler from a SCM was not used in this figure/analysis. We make the assumption that GMT change is linear in this figure (Figure 6), and compare the local temperature change when the GMT change is 1°C to the generated patterns. For the patterns, without scaling, the GMT change is 1°C. Clarification of the aforementioned sentence was done in the manuscript.

Revison: Assuming that GMT trend is linear, it appears that the regression pattern scaling method underestimates the relationship between global temperature and local temperature

when GMT change is 1ºC. However, the degree to which it overestimates the relationship is small ($< 0.08$ºC).

Comment: Page 7, line 12–13: The nonlinear evolution of temperature is related to the retreat of sea ice.

Response: Yes, thank you. We have edited this sentence.

Revision: These differences at high latitudes result in part from the nonlinear evolution of temperature due to retreating sea ice.

Comment: Page 7, line 16: R2 evidently refers to the square of correlation coefficient; this should be mentioned explicitly. Is the correlation calculated with respect to time?

Response: The $R^2$ is the square of correlation coefficient, and it is between the GMT and local temperature with respect to time. This sentence has been edited to clarify this ambiguity.

Comment: Page 7, line 20: "despite lower local and global trends". This might be removed. The ratio may well be large regardless of the magnitude of the numerator and denominator.

Response: Agreed. The aforementioned phrase has been removed.

Comment: Page 7, line 21: Over oceans in winter, the rate of warming is large before the ice melt. After the melt, the large thermal inertia of the sea water tends to decelerate warming.

Response: Yes, thank you. The sentence in question has been edited due to a prior comment, and misleading/incorrect statements have been taken out. Revision to this sentence is listed above.

Comment: Page 9, line 3: I studied one file by the ncdump command, and in that file the period for historical climatology was 1961–1990.

Response: Yes, thank you for pointing that out. The correct climatology is 1961-1990. This has been corrected in the manuscript.

Comment: Page 8, lines 13–14: "particularly when strong mitigation is employed later in the simulation". The idea is not clear for me.

Response: Yes, this phrase is confusing and not needed. It has been taken out.

Comment: In the Figures, colour hues are very close to one another, and thus it is hard to infer the actual values of the field at some specified location. For the largest values, it

would be good to use some other colours than red and blue. Alternatively, you might add some contours to facilitate interpretation.

Response: For the figures in this analysis we want to show the gradual gradient, and we believe that the red/blue color scale used for many of the figures is appropriate. We acknowledge that small differences between neighboring values may not be readily discernible, but large differences are discernible, which is our main purpose.

Comment: In Figs. 4 and 6, some other colour than white should be used to emphasize areas with statistically significant differences.

Response: Good point. Areas where there are significant differences are now emphasized using hatching.

Comment: Fig. 3: Is the variance calculated with respect to time? Standard deviation might be more illustrative than the variance. Moreover, the variability of temperature is very different in the different parts of the world. Therefore, it would be relevant to give differences in std in a relative sense (e.g., $(\sigma 1 - \sigma 2)/\sigma 2$). This would reveal where the differences actually are pronounced. After seeing the differences in rms in relative terms, it is possible to evaluate the statement "Differences in variance across epochs were also small" (page 5, line 19) and whether "these relatively small differences in variance (or std?) between epochs were not likely to affect the resulting temperature patterns".

Response: Yes, this was not clear in the text or in the corresponding figure. Because we do not want the epoch choice and the delta method patterns to be the focus of this paper, despite compelling and interesting issues brought about by Hawkins and Sutton, 2015, we have limited the discussion of epochs and the delta method. Figure 3 has been removed to streamline our discussion.

**Minor Comments:**

Comment: Page 2, line 13: "adaptability of additional predictors" is difficult to understand.

Response: Yes, this phrase was not correct. This phrase should read "adaptability to additional predictors". This has been corrected.

Comment: Page 2, line 20: Does "... vary between scenarios and for other climate variables...." mean "... vary between scenarios and climate variables"?

Response: Yes, we would agree that this line could be clearer. This sentence has been edited by deleting the "for other".

Comment: Page 2, line 31–33: "In linear regression, only the error term is assumed to have a normal distribution, so it is likely that climate extremes would yield high error terms." Please explain this more explicitly.

Response: This sentence is unclear. This line has been revised for clarification.

Revision: In linear regression, the error term is assumed to have a normal distribution with a mean of "0". So it is likely that outliers or climate extremes would be in the very end of the tails and yield high error terms.

Comment: Page 4, line 12: "a 30-year reference epoch of 1986–2005". Only 20 years!

Response: Yes, thank you. It has been corrected.

Comment: Page 4, line 15–17: The sentence should be rephrased.

Response: Agreed.

Revision: In adaptation/mitigation analyses, a pre-industrial control simulation epoch is often used as the baseline from which change is diagnosed, as this period has little to no anthropogenic forcing. However, for pattern generation, an epoch in the later half of the 20th Century is often used (Osborn, 2009; Tebaldi et al, 2014).

Comment: Page 4, line 18: 1861–1990 should be 1861–1890?

Response: Yes, good catch. Thank you, this has been corrected.

Comment: Page 5, line 2: "ensemble variance". Variance within or between the ensembles? Please specify.

Response: Yes, this was not clear in the text or in the corresponding figure. This sentence was about the ensemble mean year to year variance differences. This sentence (and Figure 3) has been removed.

Comment: Page 5, line 22: "in the observed period". You are dealing with model simulations rather than observations.

Response: Yes, this is a good point. "Observed" should be "historical." This has been corrected.

Comment: Page 7, line 6: Should the unit be ◦C/◦C rather than ◦C?

Response: Yes, thank you. This has been corrected.

Comment: Page 8, line 6: I did not understand "The local to global fit is strong".

Response: "Strong" was not the appropriate word to use. We have changed it to "good".

Comment: Page 8, line 8: Give some examples of potential additional predictors.

Response: Excellent suggestion.

Revision: "…allows for additional predictors (e.g. land/ocean difference, latitude, non-$CO_2$ aerosols) in…"

Comment: Page 8, line 16: "epoch trends"?

Response: Thank you, "trends" should be "means". This has been corrected in the text.

Comment: Page 8, line 26: Please define GitHub.

Response: Yes, good point. A sentence describing GitHub has been added to this section.

Comment: Page9,line6: 165kBto1MB.

Response: Thank you for pointing out that error. It has been corrected.

Comment: In the pdf file of the manuscript, list of references is given twice.

Response: Yes, thank you. It has been corrected.

Comment: In Table 2, are the rms differences global?

Response: Yes. This has been edited to clarify the text.

Comment: In the caption of Fig. 4, specify the RCP scenario.

Response: Good point. It has been edited.

Comment: Fig. 7: The unit is different for change ($\circ$C) and trend ($\circ$C/year).

Response: Yes, this is correct. However, this figure has been removed for reasons listed above.

Comment: In Fig. 9, consider showing the standard deviation rather than variance; give the unit of the quantity in the caption.

Response: Good suggestion. The figure has been redone with the standard deviation rather than the variance, and the units for the standard deviation scalebar have been added.

Comment: Fig. 10, caption: Replace "R-squared" by "the square of the correlation coefficient".

: Yes, thank you.  It has been replaced.

Comment: Fig. 11: Is this "Ensemble mean annual average"?

Response:  Yes, thank you.  It has been corrected.

---

## Author Comment (AC2) · 12 Apr 2017

Manuscript: "An open-access CMIP5 pattern library for temperature and precipitation: Description and methodology"

Authors: Cary Lynch, Corinne Hartin, Ben Bond-Lamberty, Ben Kravitz

Journal: Earth System Science Data

Date: 12 April 2017

Response to Reviewer #2

**Major Comments:**

Comment: I had was confused by the description provided in that it discusses the application of methodology to only 12 models that met certain criteria needed for the comparison, and only the first realization. It is not clear what was done for these other models for which there are files are available at the access sites. Does the library contain results for both methods? The netCDF file contained only temperature and precipitation output, seemingly for one method only.

Response:  We agree that our sub-selection of models and analysis of patterns generated by both methodologies did not make clear what was contained in the data library.

We realized that we had omitted our reasoning as to why we chose the 12 models we did. For this analysis we did not want to use a large ensemble of models, and we also did not want to arbitrarily choose a multi-model ensemble that would over-represent particular sub-models. For these reasons, we developed a small set of simple performance metrics for a 10-12 model ensemble that realistically produced mean observed spatial/temporal climate.   We have added this to the manuscript.

We have also clarified that although we only analyze the results for one realization from each of 12 models in the text of the paper, in the data repository, we include patterns of all available models.  As this is a multi-part issue of clarity, edits were made to the abstract, introduction, data description, and data repository section.

Comment: The paper shows results for temperature, but the library contains precipitation. Since these two variables behave quite differently on a global and temporal scale, it would be worth showing some precipitation examples, and perhaps go less in depth about temperature.

Response:  Because the relationship between global mean temperature and local precipitation is complex and requires a more in-depth examination of pattern scaling methodologies, we have written an entirely separate manuscript discussing precipitation pattern scaling (Kravitz et al., 2017).  We reference that paper in our manuscript and have

revised some of our description to make clear why the repository contains precipitation patterns.

Comment: There should be a clearer explanation about the difference between the two access points. At the doi location (mentioned in the abstract), one can download the entire packaged zipped. Once unzipped, the README is hard to open because it contains scripting language so the laptop software refuses to open it. Only annual patterns are found here. In the github location (mentioned in the section 5), one can download only single files, but the README displays easily. Once downloaded, NetCDF files from both locations open easily. Also, one site appears to have only annual files.

Response: We agree. This was due to a synchronization mistake, which has been corrected. Data from both access points are now the same.

Comment: Rather than merely listing the variables, it would have been more useful to a potential user to get a brief recap of what the variables represent or refer back to the appropriate section. For example, the grid size depends on the original model grid. Why not refer back to Table 1 for the 12 models? What about the other models? Also it was not clear if the patterns were derived on the coarse grid and then remapped, because the comparison was done on a common grid. Where is the description of the analysis done on native grid?

Response: We have edited the Pattern Library section where native model resolution is discussed to clarify that no regridding in the pattern library has been done, and added a reference to Table 1 in the section about regridding.

Comment: I cannot validate this product. The closest thing to a "validation" for this type of product might be a computation of the scaling patterns for the last 20-50 years, using the decades prior to that as the baseline, i.e., an L20C/M20C. In this way, the user can see how a model captured the changes in the last few decades, and compare with actual observed trends. This might help them users decide which model(s) might be more appropriate for this use. If this work has been done by others, a brief review of that literature would be very helpful and should be included in the paper. In re-reading this paper, it appears that the Data and results might actually represent the Validation of the dataset.

Response: We thank the reviewer for the insightful comment, and we agree that the data and results were how we validated the data set

Comment: I would also suggest a polar projected example (Arctic) for temperature, at least.

Response: Excellent suggestion. We have recreated Figure 5 with the polar projection (for the Northern Hemisphere, see figure below), and we do not believe at this point that it adds new information. We do note that there is some distortion at high latitudes with a Robinson projection, a problem which is addressed with a polar projection.

[Figure]

*Figure 5.* Ensemble mean regression method pattern and delta method pattern differences in °C/°C for L21C (2071-2100)/ L19C (1861-1890) and L21C/ L20C (1971-2000) for annual, DJF and JJA for future forcing scenario rcp8.5. Significance values below the 95% confidence interval using a Student's t-distribution probability statistic were masked.

Comment: The dataset is not complete in the sense that it only analyzes the for each model, it only analyzed one run, as stated in the paper. Thus, the library has limited value than if they analyzed all runs. Not all users will want the results for only one run, so they will have to re-compute the patterns anyway.

Response: The issue of multiple realizations is complicated, especially considering that only 14 (out of 41) modeling centers released more than 1 realization. For the high forcing scenario, we found that the differences in patterns across realizations are not statistically significant, and the patterns were highly correlated, so a single realization provides sufficient information. This may not be true for lower forcing scenarios, so if we provide patterns for those lower forcing scenarios in the future, we will address this issue.

Comment: I cannot determine if the dataset is of high quality that but it is reassuring that error fields and statistics are provided.

Response: Thanks! We have supplied the error fields exactly for that purpose.

Comment: I pasted the link listed under associated files listed for the climatology variable. It did not go anywhere.

**Response:** Thanks for pointing that out. We have fixed the broken link.

**Comment:** I am surprised SI units are not used. Perhaps there is a specific reason for this? It should be explained in the manuscript.

**Response:** We opted to keep the units in °C for easier interpretation for non-climate readers (e.g., impact analysis and other applied purposes), which is one of our target audiences. We do not anticipate a great deal of difficulty in converting between °C and K for those users who wish to do so.

**Comment:** I didn't do a detailed list of the typos, which are few. However, the paper will need to be re-organized.

**Response:** We would like to thank the reviewer for their suggestions. After thoughtful consideration and many edits to the manuscript, we believe the manuscript is clearer and cleanly describes our analysis and the resulting pattern library.

**Minor comments:**

**Comment:** Page. 4 1861 -1990 ...should be 1861-1890?

**Response:** Yes, good catch. It has been corrected.

**Comment:** Introduction: Please clarify if the patterns are purely spatial, not temporal.

**Response:** Thank you. The patterns are purely spatial and this has been made clear in the Introduction.

**Comment:** Table 1. What are the units? Is it longitude X latitude or vice versa?

**Response:** Good suggestion. We have added units to this table.

**Comment:** Table 2. Need more clarity on what this represents..difference in RMS?

**Response:** Yes, we agree that this caption is vague. It is the root mean square difference between the actual and pattern predicted output. We have edited this caption.

**Revision:** Table caption: Root mean square difference between actual and pattern predicted global mean anomalies in °C/°C for each pattern methodology at the end of the 21$^{st}$ Century.

---

## Editor Comment (EC3) · M. E. Contadakis (Editor) · 13 Apr 2017

The authors have dully respond to all comments and recommendations of both reviewers and make the appropriate changes in the revised manuscript. Therefor I am in the pleasant situation to recommend the acceptance of the revised manuscript for publication in the Journal ESSD
* * *